

# High-resolution under-water laser spectrometer sensing provides new insights to methane distribution at an Arctic seepage site

Pär Jansson[1], Jack Triest[2], Roberto Grilli[2], Bénédicte Ferré[1], Anna Silyakova[1], Jürgen Mienert[1], Jérôme Chappellaz[2]

[1] CAGE Center for Arctic Gas Hydrate, Environment, and Climate, Department of Geosciences, UiT-The Arctic University of Norway, 9037, Tromsø, Norway

[2] Univ. Grenoble Alpes, CNRS, IRD, Grenoble INP, IGE, 38000 Grenoble, France

*Correspondence to*: Pär Jansson (per.g.jansson@uit.no), Roberto Grilli (roberto.grilli@cnrs.fr)

**Abstract.** Methane (CH$_4$) in marine sediments has the potential to contribute to changes in the ocean- and climate system.

Physical and biochemical processes that are difficult to quantify with current standard methods such as acoustic surveys and discrete sampling govern the distribution of dissolved CH$_4$ in oceans and lakes. Detailed observations of aquatic CH$_4$ concentrations are required for a better understanding of CH$_4$ dynamics in the water column, how it can affect lake- and ocean acidification, the chemosynthetic ecosystem, and mixing ratios of atmospheric climate gases. Here we present pioneering high-resolution in-situ measurements of dissolved CH$_4$ throughout the water column over a 400 m deep CH$_4$ seepage area at the

continental slope west of Svalbard. A new fast-response under-water membrane-inlet laser spectrometer sensor demonstrates technological advances and breakthroughs for ocean measurements. We reveal decametre-scale variations of dissolved CH$_4$ concentrations over the CH$_4$ seepage zone. Previous studies could not resolve such heterogeneity in the area, assumed smoother distribution and therefore lacked both details and insights to ongoing processes. We show good repeatability of the instrument measurements, which are also in agreement with discrete sampling. New numerical models, based on acoustically evidenced

free gas emissions from the seafloor, support the observed heterogeneity and CH$_4$ inventory. We identified sources of CH$_4$, undetectable with echosounder, and rapid diffusion of dissolved CH$_4$ away from the sources. Results from the continuous ocean laser-spectrometer measurements, supported by modelling, improve our understanding of CH$_4$ fluxes and related physical processes over Arctic CH$_4$ degassing regions.

## 1 Introduction

Methane (CH$_4$) release from gas bearing ocean sediments has been of high interest for many years (e.g. Jørgensen et al., 1990; Westbrook et al., 2009; Ferré et al., 2012; Boetius and Wenzhöfer, 2013; Myhre et al., 2016; Ruppel and Kessler, 2016; Platt et al., 2018). Once released and dissolved in the water column, the CH$_4$ gas diffuses and is partly oxidized in the water column (Reeburgh, 2007), contributing to ocean acidification (Biastoch et al., 2011) and minimum oxygen zone formation (Boetius and Wenzhöfer, 2013). Chemosynthetic life on the seabed depends on the supply of methane as an energy resource (e.g.

Boetius and Wenzhöfer, 2013). Supply of nutrient rich bottom water, by means of local upwelling, may enhance biological productivity, induce drawdown of CO$_2$ from the atmosphere, potentially making shallow CH$_4$ seepage sites sinks for this critical greenhouse gas (Pohlman et al., 2017). Warming of ocean bottom waters, active tectonics and ice sheet build up and retreat could, at different time scales, lead to CH$_4$ gas release from the seabed (e.g. Portnov et al., 2016). The magnitude and trend of such a phenomena are still under debate (e.g. Ruppel and Kessler, 2016; Andreassen et al., 2017; Hong et al., 2018)

and accurate methods to measure methane concentration from its source are needed. At shallow seepage sites, such as the East Siberian Arctic Shelf, CH$_4$ can potentially reach the atmosphere and amplify greenhouse warming (Shakhova et al., 2010; Shakhova et al., 2014). However, most studies of shallow CH$_4$ seepage sites have found no or little CH$_4$ flux to the atmosphere (e.g. Gentz et al., 2014; Myhre et al., 2016; Miller et al., 2017; Platt et al., 2018).



In the past, most $CH_4$ measurements relied on indirect or discrete sample measurements (e.g. Damm et al., 2005; Westbrook

et al., 2009; Gentz et al., 2014). Bubble catcher and mapping with multibeam echosounder (Sahling et al., 2014), hydro-

acoustic imaging together with bubble size and bubble rising speed measurements (Ostrovsky, 2003; Greinert et al., 2006;

Sahling et al., 2014; Weber et al., 2014; Veloso et al., 2015) have been used to derive $CH_4$ flow rates. The acoustic indirect

method can only quantify $CH_4$ fluxes from acoustically detectable bubbles, and ROV's can only capture visible bubbles, while

neither can detect $CH_4$ from sources other than free gas seepage. These methods do not provide information about the

distribution of dissolved $CH_4$ in the water column. Discrete water sampling with Niskin bottles only allows measurement of

the dissolved $CH_4$ at limited spatial resolution, and the low horizontal and vertical resolution may lead to artificial smoothing

of the spatial distribution and inaccurate estimate of average dissolved $CH_4$ concentration. The method using the combination

of bubble catcher and multibeam echosounder introduces large uncertainties while extrapolating $CH_4$ flow rates from few

bubble catcher measurements and applying those flow rates to acoustically evidenced bubble streams (flares). Present

commercial underwater $CH_4$ sensors do not have the required response time for accurate high-resolution mapping. For this

reason, Gentz et al. (2014) deployed an underwater membrane inlet mass spectrometer (UWMS) with a fast response time for

mapping of $CH_4$ at shallow (10 m) depths. Boulart et al. (2013) used an in-situ, real time sensor in the Baltic Sea, but it was

not deployed over a $CH_4$ seepage site. Furthermore, their reported instrument response time of 1–2 minutes and the detection

limit of 3 nmol l$^{-1}$ represent limitations for fast profiling and background concentration studies linked to the atmospheric $CH_4$

mixing ratio.

Here we present the first in-situ, high-resolution ocean laser spectroscopy mapping of dissolved $CH_4$ in seawater over active

$CH_4$ seepage in the Arctic. The data was collected by deploying a patent based (Triest et al. patent France No. 17 50063)

membrane inlet laser spectrometer (MILS) (Grilli et al., 2018). The high-resolution measurements, together with echosounder

data, discrete water sampling, and newly developed control volume and 2-dimensional (2D) models improve our understanding

of $CH_4$ fluxes from the seabed into oceans and lakes, and potentially to the atmosphere.

## 2 Materials and methods

### 2.1 Study area

The survey was performed on board R/V Helmer Hanssen, UiT, The Arctic University of Norway, in October 2015 (CAGE

15-6 cruise) west of Prins Karls Forland located offshore western Svalbard. Over a period of three days (October 21–23), we

surveyed an area of ~18 km$^2$ at water depths between 350 and 420 m, using continuous under-water laser spectroscopy as well

as traditional discrete sampling for dissolved $CH_4$, and echosounding for bubble detection and gas seepage quantification. The

study area is located at 78°33´ N 9°30´ E over an active $CH_4$ venting area (Fig. 1a). Here, more than 250 flares (acoustic

signature of bubble streams in echograms) exist along the shelf break (e.g. Damm et al., 2005; Westbrook et al., 2009; Berndt

et al., 2014; Sahling et al., 2014; Graves et al., 2015). The northward flowing West Spitsbergen Current (WSC), which



transports Atlantic Water (AW, S>34.9, T>3° C) (Schauer et al., 2004), controls the hydrography of the study area. The East

Spitsbergen Current (ESC), flows south-westward along the eastern Spitsbergen coast, and northward along the western

Svalbard margin, carrying Arctic Surface Water (ASW, 34.4⩽S⩽34.9) and Polar Water (PW, S<34.4) (Skogseth et al., 2005).

The Coastal Current (CC), extension of the ESC (Loeng, 1991; Skogseth et al., 2005), contributes a transient addition of ASW

and PW on the shelf and the continental slope as the WSC meanders on- and offshore (Steinle et al., 2015). The Lower Arctic

Intermediate Water (LAIW, S>34.9‰, T⩽3 °C) flows below the Atlantic Water (Ślubowska-Woldengen et al., 2007).

### 2.2 Hydrocasts with discrete water sampling

Vertical oceanographic profiles were recorded at 10 stations (Fig. 1a) using a SBE 911 plus CTD (Conductivity, Temperature,

and Depth) mounted on a rosette, which carried twelve 5-liter Niskin bottles. In January 2015, the CTD was fitted with new

sensors; an SBE 4 Conductivity Sensor and an SBE 3plus Premium CTD Temperature Sensor, with initial accuracies of $\pm$

0.001 °C and $\pm$ 0.0003 S m$^{-1}$. At 24 Hz sampling, the resolutions are 0.0003 °C and 0.00004 S m$^{-1}$.

The Niskin bottles were closed during the up-casts, collecting seawater at different depths for further dissolved $CH_4$ analysis.

Headspace equilibration followed by gas chromatography (GC) analysis was carried out in the laboratory at the Department

of Geoscience at UiT, The Arctic University of Norway, using the same technique as Grilli et al. (2018). The resulting

headspace mixing ratios (ppmv) were converted to in-situ concentrations (nmol l$^{-1}$), using Henry's solubility law, with

coefficients calculated accordingly with Wiesenburg and Guinasso (1979). The sample dilution from addition of a reaction

stopper (1 ml of 1M NaOH solution replacing 1 ml of each 120 ml sample), and the removal of sample water while introducing

headspace gas (5 ml of pure $N_2$ replacing 5 ml of sample water) was accounted for. The overall error for the headspace GC

method was 4%, based on standard deviation of replicates.

### 2.3 Methodology and technology for high-resolution laser spectrometer $CH_4$ sensing

A stainless steel frame attached to a cable that was connected to an on-board winch served as a platform to which the MILS,

an Aanderaa, Seaguard TD262a CTD, a standard commercial $CH_4$ sensor, and a battery pack were mounted. This instrument

assembly, hereafter called the probe, has a total height of ~1.8 m, a total weight in air of ~160 kg and a negative buoyancy of

~52 kg. We towed the probe for a total of 28 hours, providing unsurpassed high-resolution in-situ $CH_4$ measurements with a

sampling rate of 1 s$^{-1}$, together with dissolved oxygen data, as well as pressure, temperature and salinity. The sensors fitted to

the Aanderaa CTD, a Conductivity Sensor 4319, a Temperature Sensor 4060, and an Oxygen Optode 4330, has initial

accuracies of $\pm$ 0.005 S m$^{-1}$, $\pm$ 0.03°C, and < $\pm$ 8 µM and the resolutions are 0.0002 S m$^{-1}$, 0.001 °C, and < 1 µM,

respectively.

Lowering and heaving of the probe in the water column allowed for vertical casts, while towing the probe behind the moving

ship at varying heights above the seafloor generated near-horizontal trajectories. The main horizontal trajectories comprise

five lines (Fig. 1a), where the desired distance (~15 m) from the seafloor was attained by monitoring the pressure in real time



while adjusting the cable payout. The battery-powered MILS (Fig. 1b, see Grilli et al. (2018) for more details) has a membrane

inlet system, linked to an optical feedback cavity-enhanced absorption spectrometer and an integrated PC for control and data

storage. Cabled real-time communication with the instruments allowed instant decision-making, and ensuring optimal sensor

operation during the deployments.

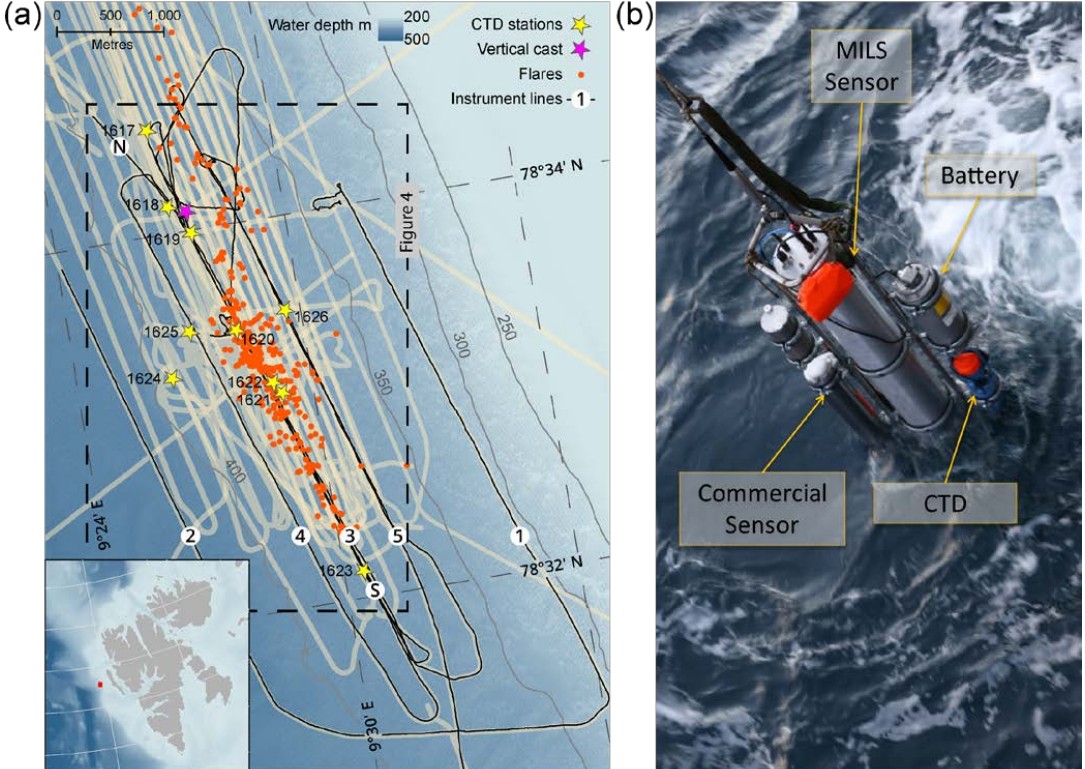

Figure 1: Map of the surveyed area and photo of the instrument assembly. a) Survey lines and sampling locations over the study
area at the Svalbard continental margin. Black lines show the ship trajectory with line numbers assigned in the order they were
surveyed. Beige areas appearing as thick lines indicate echosounder beam coverage from this campaign and previous cruises (AOEM
2010, and CAGE 13-7). The start- and end- locations of line 3 are indicated with N and S respectively. Known flare locations from
this survey and surveys in 2010 and 2013 are marked with orange dots. CTD stations with discrete water sampling are marked with
yellow stars and the vertical instrument cast with a purple star. The inset image shows an overview of Svalbard with the survey
location indicated with a red square. b) Instrument assembly. The main central tube is the prototype MILS sensor. The stainless
steel frame acts as a platform and allows attachment of instrument battery (top right side), CTD (blue at the bottom right) and a
commercial $CH_4$ sensor and its battery pack (left side).

Sensors with membrane inlets can be sensitive to fluctuating water flow over the membrane, which can result in artificial

variability of measured concentrations. Careful positioning of the Sea-Bird SBE5T water pump to minimize inlet, and outlet

pressure changes and subsequent flow variations minimized this effect. The water pump inlet has a fine mesh filter and a shield

to avoid entry of free gas bubbles and artefacts from gas bubbles entering the sampling unit and reaching the membrane surface.

All parameters from the MILS sensor, including gas flow, pressure, sample humidity, and internal temperature were logged to

process and evaluate the quality of the data. A dedicated ship-mounted GPS logged positional data for accurate synchronization





of the probe and ship position. A position correction, accounting for the lag between the probe and the ship synchronizes the

towed instrument data with simultaneously acquired echosounder data. The Matlab routine "Mooring Design and Dynamics"

(Dewey, 1999) simulated the towing scenario, for which we used a simplified instrument assembly composed by a cylinder

1.68 m long, 0.28 m diameter with a negative buoyancy of 52 kg. A polynomial speed-factor ($x_* = -0.2211u^5 + 1.355u^4 -$

$3.0126u^3 + 2.6741u^2 - 0.1609u$) was derived to account for the combined ship- and water current velocities ($u$ in m s$^{-1}$).

The distance of the probe behind the ship and the corresponding required time-shift was calculated by multiplying the non-

dimensional speed-factor ($x_*$) with the instrument depth at each data point. This approach allowed for dynamic correction of

data positions, accounting for towing with or against the water current and a near-stationary ship during vertical profiling.

Correction for tidal currents was neglected since tides constituted less than 5% of the WSC of ~0.2 m s$^{-1}$ during our

deployments, according to the tide model TPXO (Egbert and Erofeeva, 2002).

Mixing ratios of CH$_4$ (ppmv) measured by the MILS were converted into aqueous concentrations (nmol l$^{-1}$) using Henry's law,

where the solubility coefficients were determined accordingly with Wiesenburg and Guinasso (1979), while accounting for in-

situ pressure, temperature, and salinity. The uncertainty of the dissolved CH$_4$, measured with the MILS is ± 12% (Grilli et al.,

2018).

**2.4 Acoustic mapping and quantification of seafloor CH$_4$ emissions**

Gas bubbles in the water column are efficient sound scatterers and ship-mounted echosounders can therefore be used for

identifying and quantifying gas emissions (Ostrovsky et al., 2008; Weber et al., 2014; Veloso et al., 2015). The target strength

(TS), defined as 10 times the 10-base logarithmic measurements of the frequency dependent acoustic cross sections (Medwin

and Clay, 1997), quantifies the existence of sound scattering objects in the water column. Time series of TS are displayed in

so-called echograms (Greinert et al., 2006; Judd and Hovland, 2009). During the cruise, the 38 kHz channel of the ship-

mounted single beam Simrad EK-60 echosounder recorded acoustic backscatter continuously. Flares can be identified in the

echograms and distinguished from other acoustic scatter from fish schools, dense plankton aggregations, and strong water

density gradients. We identify flares as features in echograms, which exceed the background backscatter (TS) by more than

10 dB, with a vertical extension larger than their horizontal, and which are attached to the seafloor.

We used the methodology developed and corrected by Veloso et al. (2019) and the prescribed FlareHunter software for

145 mapping and quantifying gas release. For the flow rate calculations performed with the Flare Flow Module of FlareHunter, we

used the bubble size distribution previously observed in the area (Veloso et al., 2015). Temperature, salinity, pressure, and

sound velocities, all required for correct quantification, were provided by the CTD casts. The resulting flow rates and seepage

positions allow for mass balance calculation in the control volume model and in the two-dimensional (2D) model, as described

in Sect. 2.5 and 2.6, respectively.



### 2.5 Control volume model

The temporal evolution $(dC/dt)$ of a solute's concentration $C$ within a certain volume $V$, with water flowing through it can, using mass conservation, be written as:

$$\frac{dC}{dt} = \frac{Q_{IN} \times C_B}{V} - \frac{Q_{OUT} \times C}{V} + \frac{F}{V} + k\nabla^2 C \tag{1}$$

Equation (1) is a second order differential equation, from which an analytical steady state solution can be derived by following these assumptions: The volumetric flow of water in and out of the control volume, $Q_{IN}$ and $Q_{OUT}$ are balanced and are given by a steady water current in the x-direction across the width ($\Delta y$) and height ($\Delta z$) of the control volume. The diffusion is kept homogenous and constant by applying a constant diffusion coefficient $k$. The background concentration $C_B$ is fixed in time and space and $F$ represents the persistent flow of the solute (in this case bubble mediated $CH_4$) into the volume. The $CH_4$ dissolves completely within the volume, and the diffusion occurs across the domain (in the y-direction). Using the central difference approximation of the second derivative ($\nabla^2$ in Eq. (1) and the above assumptions yield that the aqueous $CH_4$ within the volume reaches the steady state concentration:

$$C_{t=\infty} = \left( \frac{Q_{IN} \times C_B}{V} + \frac{F}{V} + \frac{2k \times C_B}{(\Delta y)^2} \right) \times \left( \frac{Q_{OUT}}{V} + \frac{2k}{(\Delta y)^2} \right)^{-1} \tag{2}$$

Finally, by averaging measured $CH_4$ concentration within a defined volume, and assuming that it represents a steady state concentration, the bubble flow rate is retrieved from Eq. (2).

$$F = (\bar{C} - C_B) \times \left( Q + \frac{V \times 2k}{(\Delta y)^2} \right) \tag{3}$$

Where $\bar{C}$ represents the measured average concentration, and $Q = Q_{IN} = Q_{OUT}$.

The dimensions of the control volume with volume $V = \Delta x \times \Delta y \times \Delta z$, were chosen to match the length of line 3 ($\Delta x = 4.5$ km), extended 25 m perpendicularly on each side of the line ($\Delta y = 50$ m), and 75 m vertically ($\Delta z = 75$ m).

### 2.6 Two-dimensional model

In order to gain insight to the physical processes behind the observed $CH_4$ variability, we constructed a two-dimensional (2D) numerical model, resolving the evolution of dissolved $CH_4$ in the water column, which results from $CH_4$-bubble emissions, advection with water currents and diffusion. The model domain was made 400 m high in the z-direction, 4.5 km long in the x-direction, and oriented along line 3 (Fig. 1a). The navigation data along this line is linearly interpolated to form the basis for a 2-metre gridded model domain starting at 78°34.54'N 9°25.92'E and ending at 78°32.1' N 9°30.58' E as indicated by N and S, in Fig. 1a. FlareHunter derived flow rates within 50 m from line 3 were projected into the model domain, and the source of dissolved $CH_4$, mediated by bubbles, was distributed vertically by applying a non-dimensional source-function similar to the approach of Jansson et al. (2019): $S0(z) = 6.6 \times 10^{-2} \times e^{-0.066 \times z}$, where z is the vertical distance from the seafloor in metres. We calculated source distribution functions $S(z)$ by scaling $S0(z)$ with the flare flow rates, and distributed the resulting source into current-corrected x/z nodes with volumes $\delta V = \delta x \times \delta y \times \delta z$, where $\delta x = \delta y = \delta z = 2$ m. The model domain comprises 12 extra cells on each side in the y-direction in order to avoid fast diffusion out of the domain while the background concentration



is held constant. The 2D model simulated $CH_4$ diffusion and advection with water currents, and was run to steady state using

different diffusion coefficients, within the range suggested by Sundermeyer and Ledwell (2001).

## 3 Results

### 3.1 Water properties

The water is well mixed within 150 masf (metres above the seafloor) and continuously stratified from 250 to 50 mbsl (metres

below the sea level) (Fig. 2a) with a squared buoyancy frequency of $\sim N^2 < 4\times10^{-5}$ $s^{-2}$. A pycnocline exists at $\sim$30 mbsl (Fig.

2a) with $N^2$ up to $10^{-4}$ $s^{-2}$, marking the transition between surface water and AW below (Fig. 2b and 2c). Temperatures close

to the seafloor range from 4.2–4.4 °C, which is more than 1 °C above the $CH_4$ hydrate stability limit (Tishchenko et al., 2005),

for a salinity of 35.1 as indicated in Fig. 2a. The velocity of the WSC was between 0.1 and 0.3 m $s^{-1}$ (Fig. 2d) inferred from

the inclination of flare spines (Veloso et al., 2015), and followed the isobaths, which is consistent with previous findings (Gentz

et al., 2014; Graves et al., 2015). The mean salinity and temperature, in different layers, with their corresponding standard

deviations according to the water masses classification of Skogseth et al. (2005) and Ślubowska-Woldengen et al. (2007) are

shown in Fig. 2b,c. The temperature/salinity distribution suggests a clear dominance of AW during the survey, overlaid with

fresher and colder ASW and PW.






Figure 2. Hydrography during the survey. a) CTD casts 1617- 1626 showing temperature (red), salinity (blue) and density anomaly (green) calculated with Gibbs sweater package (McDougall and Barker, 2011). b) Temperature and salinity (TS) diagram coloured by pressure (dbar). Grey curved lines in the background indicate isopycnals (constant density ($\sigma$) lines). AW indicates Atlantic Water, PW is polar water, ASW is arctic surface water and LAIW is Lower Arctic Intermediate Water. Water mass definitions are described in the text. Black dots indicate the mean water properties for the different layers and crosses indicate the corresponding standard deviations. c) TS diagram coloured by CH$_4$ concentrations (nmol l-1) measured with the MILS. Black dots depict average temperature and salinity at water depth intervals, and the error bars indicate the corresponding standard deviations. d) Water currents inferred from inclination of flare spines (Veloso et al., 2015) with a mean bubble rising speed of 23 cm s-1.







### 3.2 Measured and modelled CH₄ distribution

The high-resolution dissolved $CH_4$ concentration profiles resulting from towing the MILS along five lines, approximately 15 masf show high variability (Fig. 3), especially over line 3, which geographically matches the clustering of bubble plumes (Fig. 1a).

On the landward side (lines 1 and 5), the concentration is relatively smooth with an average of ~55 nmol $l^{-1}$, but along line 5, which is closer to the main seepage area, the concentration is influenced by the nearby seepage, inferred from the concentration peaks reaching up to 105 nmol $l^{-1}$ at 78°33.5' N. On the offshore side, the mean concentrations are 15 and 36 nmol $l^{-1}$ along lines 4 and 2, respectively with elevated $CH_4$ concentrations of up to ~ 70 nmol $l^{-1}$, lacking hydroacoustic evidence of $CH_4$ seep sources. The peak in line 4 may be explained by its proximity to the main bubble seep cluster, but the $CH_4$ concentrations

show more variability along line 2, the offshore-most horizontal trajectory of the survey, which may indicate undetected $CH_4$ seepage located deeper than 400 mbsl.

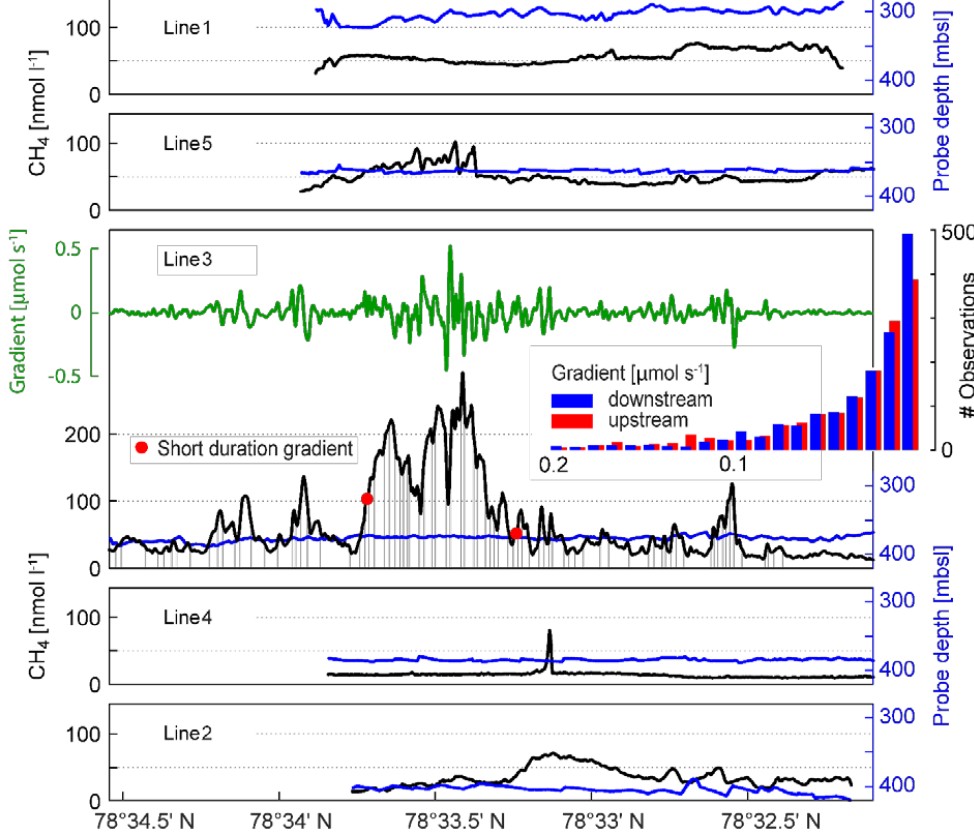

**Figure 3.** MILS measurements along the five lines ~15 m from the seafloor. Panels show data acquired along lines 1–5, shown in order of proximity to the shore, with line 1 closest to the shore and line 2 furthest offshore. See Fig. 1a for line locations. Black lines
show $CH_4$ concentrations, blue lines show the probe depth. The concentration gradient along line 3 is shown with a green line and the red- and blue bar chart indicates its probability distribution. Grey vertical lines indicate the slopes chosen for calculation of the mean gradients away from the $CH_4$ sources. Red dots indicate instances where concentrations changed during periods shorter than the MILS response time and thus, where the concentration gradients are possibly limited by the instrument response time.





A 25 minute down- and upward sequence obtained from the vertical MILS cast at station 1616 (Fig. 4), shows excellent

repeatability after correcting for an instrument time lag of 15 seconds, representing the time required for the gas mixture to

reach the measurement cell. The sensor showed no memory effects, i.e. different response time between increased and

decreased $CH_4$ concentrations.

Analysis of discrete samples (DS) from CTD casts 1618 and 1619 and the vertical MILS cast 1616 give further insights to the

heterogeneity and temporal variation of the dissolved $CH_4$ distribution (Fig. 4). Discrete measurements from CTDs 1618 and

1619 reveal a qualitative match with the MILS measured concentrations extracted from line 3 near these stations (red and

green symbols in Fig. 4). Discrepancies between the MILS cast 1616 and the DS from CTDs 1618 and 1619 close to the

seafloor is likely due to the difference in sampling location, as the MILS vertical cast 1616 was ~150 and ~180 m away from

CTDs 1618 and 1619, respectively.

The exponential "dissolution" function, which represents the expected trace of dissolved $CH_4$ in the water column, resulting

from bubble dissolution, was compared to the entire MILS dataset by plotting $CH_4$ concentrations against height above the

seafloor, determined from position corrected pressure and previously acquired multibeam data (Fig. 4).

Elevated $CH_4$ concentrations at ~160 and ~220 mbsl revealed by the MILS vertical profile 1616 was not identified with DS

from the nearby CTD cast 1619, and DS from CTD 1618 reveal only a small fraction of the $CH_4$ anomaly, because of too

sparse sampling (Fig. 4). The MILS data collected 15 masf along line 3 reveals 50 nmol $CH_4$ $l^{-1}$ while the vertical profile only

30 metres away (MILS-cast 1616), measured ~200 nmol $CH_4$ $l^{-1}$ (Fig. 4). This emphasizes the strong spatiotemporal variability

of the $CH_4$ distribution in the area.

Despite the high $CH_4$ variability in the horizontal profiles (Fig. 3), further analysis of the data may be obtained by focusing on

line 3, towed in north-south direction at ~0.8 m $s^{-1}$ directly over the bubble streams. The fast response time of the MILS sensor

($T_{90}$ = 15 s) revealed decametre-scale variations of the dissolved $CH_4$ concentrations with high values well correlated with the

echosounder signal, after correcting for the towed instrument position (Fig. 5).



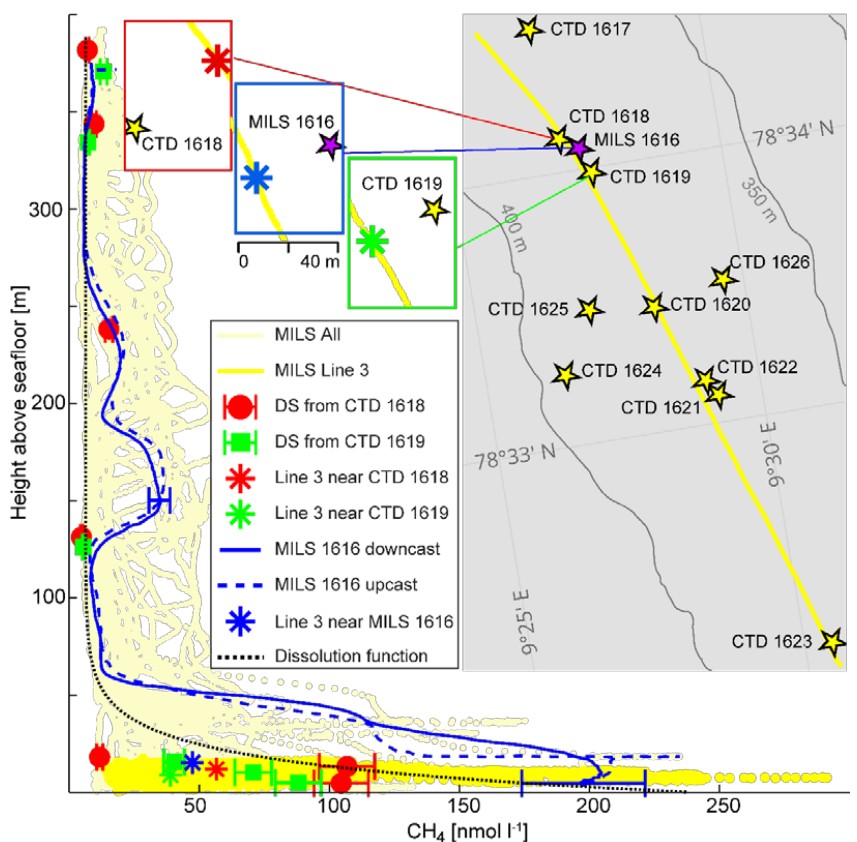

**Figure 4.** High-resolution CH$_4$ concentrations and discrete samples. Light yellow lines show CH$_4$ concentrations acquired with the MILS during the entire survey and the bright yellow derives from line 3 at ~15 masf only. Solid and dashed blue lines represent continuous down- and upward profiles acquired at station 1616 after correction for instrument response time. The blue error bars indicate the instrument uncertainty of 12%. Discrete sample data is shown as red dots (CTD 1618) and green squares (CTD 1619) with error bars that indicate the discrete sampling/ headspace GC method uncertainty of 4%. The asterisks indicate MILS data points from the towing along line 3, closest to the vertical cast 1616 (blue), to CTD 1618 (red) and CTD 1619 (green). The black dotted line indicates the exponential dissolution function described in the text. The inset map shows the locations of the CTDs with discrete sampling (stnr1617–1623) (yellow stars) as well as line 3, which is indicated with a yellow line. The blue rectangle shows the location of the vertical MILS profile from stnr1616 (purple star) and the data point from line 3, which is closest to the deepest location of the vertical cast (blue asterisk). The green rectangle shows the location of CTD 1619 and the closest point on line 3 (green asterisk), while the red rectangle shows the location of CTD 1618 and the corresponding point on line 3 (red asterisk).

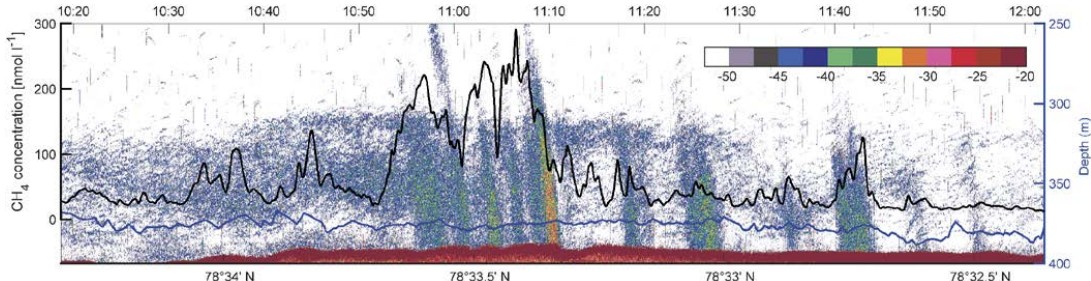

**Figure 5.** Towed MILS data overlying echo-sounder data. The black line shows the CH$_4$ concentration along line 3 (see Fig. 1 for location) at ~15 m from the seafloor. The blue line indicates the depth of the probe. The echogram, displaying TS values (colour bar shows intensity (-dB)) from the 38 kHz-channel of the EK60, is shown in the background.





A close analysis of the measured concentration reveals that the up- and down- stream gradients are equally distributed (bar chart in Fig. 3). This symmetry suggests that $CH_4$ disperses fast and equally in all horizontal directions around the bubble

plumes while being advected away from the source.

The measured $CH_4$ concentrations along line 3 changed significantly (5% or more) on sub-response times (<15 s) in only two instances and over a total time of 26 s, out of 1h 42 min, as indicated with red dots in Fig. 3. This suggests that the MILS resolved 99.6% of the gradients and that the response time of the MILS did not limit the resolution of the $CH_4$ distribution. The mean absolute gradient, assessed from steadily increasing or decreasing concentrations (grey vertical lines in Fig. 3 show

the position of the selected slopes), was 1.5 nmol $l^{-1}$ $m^{-1}$, corresponding to 1.2 nmol $l^{-1}$ $s^{-1}$. The minimum and maximum lateral gradients were -5.0, and 4.8 nmol $l^{-1}$ $m^{-1}$, which corresponds to -4.1 and 4.6 nmol $l^{-1}$ $s^{-1}$. Correlations of $CH_4$ concentrations versus depth and speed changes were low (R= 0.0133, -0.0001, -0.0094, 0.0028 for ship speed, ship acceleration, vertical instrument speed and instrument acceleration, respectively), showing the stability of the instrument during rapid movements and disproving artefacts due to water flow fluctuations at the membrane.

Sources of $CH_4$ constraining the control volume and 2D model were obtained from the acoustic mapping and quantification described in section 2.4. During the entire survey, we identified 68 unique groups of bubble plumes, with an average flow rate of 48 (SD = 50) ml $min^{-1}$. Within 50 metres of line 3, we acoustically identified 31 flares with an average flow rate of 60 (SD = 65) ml $min^{-1}$ amounting to a total flow rate of 1.87 l $min^{-1}$. These flow rates were taken as sources in the control volume and 2D model. Flarehunter calculates the flow rates in a layer 5–10 m above the seafloor. In order to calculate flow rates from the

seafloor, we upscaled the Flarehunter flow rates by 40% to compensate for bubble dissolution near the seafloor, in accordance with the dissolution profile.

The 2D model was run to steady state with different diffusion coefficients, $k \in [0.3 - 4.9 \text{ m}^2 \text{ s}^{-1}]$, adopted from dye-experiments offshore Rhode Island (Sundermeyer and Ledwell, 2001). These coefficients are in agreement with the ones obtained from the Celtic Sea ($k \in [0.8 - 4.4 \text{ m}^2 \text{ s}^{-1}]$) (Stashchuk et al., 2014), but much higher than the coefficient applied

by Graves et al. (2015) ($k = 0.07 \text{ m}^2 \text{ s}^{-1}$). The best fit between the 2D model and the MILS data (R = 0.68) was achieved during a simulation with $k = 1.5$ $m^2$ $s^{-1}$. Because the high-end coefficients of Sundermeyer and Ledwell (2001) and Stashchuk et al. (2014) were derived during wavy conditions, and because our model mainly resolves the near-bottom region, away from wave action, we interpret that our best-fit diffusion coefficient is relatively high. The resulting range of model outputs and the best fit-model simulation are visualized and compared with high-resolution measurements in Fig. 6. Despite applying a high

diffusion coefficient, the 2D model shows a residual downstream tailing, which is not seen in the MILS data. We attribute this to the fact that the model does not resolve small scale eddies, but only diffusion across the domain and diffusion/advection along the domain.



The salinity and temperature profiles of the towed CTD indicate well-mixed water, particularly over the most prominent gas

flares. Here, the standard deviation of the temperature and salinity drops by a factor of four, as highlighted by the dashed-line

box in Fig. 6. We interpret that this is caused by turbulent mixing enhanced by the bubble streams.

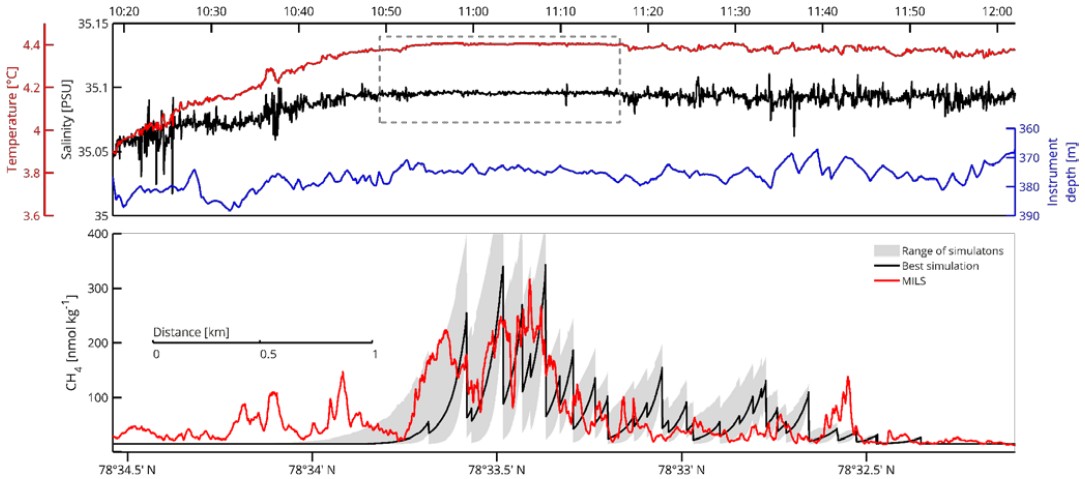

**Figure 6. Water properties and comparison between modelled and measured dissolved CH₄ concentrations along line 3. Top panel shows temperature and salinity data together with the depth of the towed instruments. The dashed-line box highlights the area of intense mixing. In the lower panel, the red line shows the dissolved CH₄ measured by the MILS. The grey area indicates the range**
**of CH₄ concentrations from the 2D model simulations. The black line depicts output of the model simulation with the best match with the measured concentrations.**

### 3.3  Methane inventory

The method, dimensions, and resolution chosen for calculating CH₄ inventories may strongly influence the resulting content

and average concentrations. This may have serious implications when the results are used for upscaling. To highlight this, we

applied different inventory calculation methods on the same water volume.

Averages along line 3 were calculated from: a) Concentrations from discrete sampling, based on different sampling depths. b)

Discrete data from different depths, linearly interpolated along the line. c) High-resolution data obtained from the MILS data

~15 masf. d) Concentrations extracted from the 2D model output at steady state at 15 masf.

Average concentrations were calculated in a "box" volume equivalent to MILS line 3 (4.5 km long (x-direction), 50 m wide

(y-direction), equivalent to the echosounder beam width, 75 m high (z-direction) corresponding to the most dynamic and CH₄

enriched zone (e.g. McGinnis et al., 2006; Graves et al., 2015; Jansson et al., 2019). Box averages were derived as follows:

The volume was divided into 1 m cubic cells. Cells located in the y-centre and in z-positions vertically matching the underlying

data (DS or MILS) were populated with the MILS, or interpolated DS profiles. The remaining cells were populated by

perpendicular and vertical extrapolation following the typical horizontal gradient of 1.5 nmol l⁻¹ m⁻¹, and vertical dissolution

profiles scaled by the measured or interpolated concentrations. The mean concentrations from the 2D model was delimited by

the height of the box. The control volume model provided only one value for the entire box.

The underlying data and its interpolation is seen in Fig. 7 and the resulting averages are reported in Table 1.



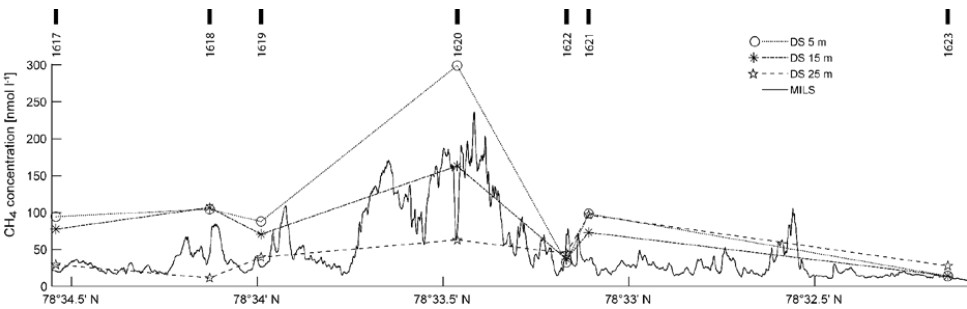

**Figure 7: Underlying CH₄ concentration data for inventory calculations. Solid line shows the continuous MILS profile ~15 masf along line 3. DS concentrations at various depths are shown as circles, asterisks and stars for 5, 15 and 25 masf respectively, and dotted, dashed, and dash-dotted lines represent the corresponding linear horizontal interpolations. CTD cast numbers are marked with thick lines at the top of the graph.**

The average $CH_4$ concentration in the box volume based on continuous data is similar to the average obtained from discrete

data at 15 m above the seafloor. We obtained 47 vs 77 nmol $l^{-1}$ for the high-resolution line and the interpolated DS, while the

box averages for the high-resolution and interpolated DS were 22 vs 29 nmol $l^{-1}$. The 2D model yielded a line average of 60

nmol $l^{-1}$, while it was 22 nmol $l^{-1}$ for the box. The control volume model predicted a steady state concentration of 23 nmol $l^{-1}$

when the diffusion coefficient of 1.5 m² s⁻¹, inferred from the 2D model was applied.

## 4 Discussion

During our survey, the mean flow rate at the seafloor per flare within 50 m of line 3, was 84 (SD = 91.6) ml min⁻¹, (min = 15.8,

max= 355.6 ml min⁻¹). This is comparable with the flow rate per flare of 125 ml min⁻¹, estimated by Sahling et al. (2014) who

assumed that an acoustic flare consists of 6 bubble streams, each with a flow rate of 20.9 ml min⁻¹. The authors found 452

flares in the area, for which they assumed similar flow rates, and thereby calculated a total flow in the area of ~57 l min⁻¹. Our

study encompasses a smaller area, where we only detected 68 flares (31 flares within 50 m of line 3) and the total flow rate

from these 68 flares was 4.56 l min⁻¹. This total flow translates to 65.7 t $CH_4$ y⁻¹ assuming constant ebullition, which may be

compared to $CH_4$ seepage of ~150 t $CH_4$ y⁻¹ estimated for a larger area by Veloso et al. (2019) and ~400 t $CH_4$ y⁻¹ (Sahling et

al., 2014), in a study area covering ours, but also extending northwards, where additional gas venting occurs. A comparison of

studies from the same area, using different methods, shows a large range of yearly $CH_4$ emissions to the water column. Flow

rates of $CH_4$ per metre of the continental shelf from previous studies given by the authors (900 kg m⁻¹ y⁻¹ (Westbrook et al.,

2009); 141 kg m⁻¹ y⁻¹ (Reagan et al., 2011); 13.8 (6.9–20.6) t m⁻¹ y⁻¹ (Marín-Moreno et al., 2013); 2400 (400–4500) mol m⁻¹ y⁻¹ (Sahling et al., 2014); 748 (561–935) t m⁻¹ y⁻¹ (Berndt et al., 2014)) yield emissions of 4050, 635, 992, 173 and 54000 t $CH_4$

y⁻¹ over the 4500 m section of the continental shelf which corresponds to our line 3.

The MILS data collected 15 masf along line 3 did not reveal the high concentrations (~200 nmol $l^{-1}$) measured during the

vertical cast only 30 m away, emphasizing the heterogeneous $CH_4$ distribution, and highlighting the need for high-resolution

sensing, rather than sparse discrete sampling.



The fast response of the MILS helped revealing decametre scale variability of dissolved $CH_4$, and we conclude that uncertainties introduced by MILS response time were negligible in this survey. The observed symmetry of $CH_4$ gradients suggests fast dispersion in all horizontal directions while enriched water is advected away from the sources.

Gentz et al. (2014) and Myhre et al. (2016) suggested that a pronounced pycnocline is a prerequisite to limit the vertical
transport of dissolved $CH_4$ towards the surface. One should note that this hypothesis was based on discrete sample data, rather than high-resolution data. We observed high $CH_4$ concentrations up to 75–100 masf, which is in agreement with bubble models (e.g. McGinnis et al., 2006; Jansson et al., 2019), highlighting that bubbles of observed sizes (~3 mm average equivalent radius) are fully dissolved within this range. Density stratification plays an important role in the vertical distribution of dissolved $CH_4$ because turbulent energy is required to mix solvents across isopycnals. Vertical mixing is therefore inhibited
even without the presence of a strong pycnocline. We suggest that the observed height limit is a result of rapid bubble dissolution and inefficient vertical mixing, regardless of the existence of a pronounced pycnocline.

We observed $CH_4$ concentrations of up to 100 nmol l$^{-1}$ without the acoustic signature of flares north of the active flare zone (Fig. 5). Echograms from the CAGE 15-6 survey (this work) and previous surveys conducted in 2010 (AOEM 2010 cruise, University of Tromsø, with R/V Jan Mayen) and 2013 (CAGE 13-7 cruise, with RV Helmer Hanssen (e.g. Portnov et al.,
2016)) reveal that the nearest bubble stream is located ~300 m northeast of this $CH_4$ anomaly. Several hypotheses may explain this $CH_4$ enrichment: a) nearby presence of $CH_4$ enriched water seepage (hypothetically from dissociating hydrates) from the seafloor; b) presence of bubble streams with bubbles too small to be detected by the echosounder (the detection limit (TS<-60 dB) of a single bubble was 0.42 mm for this survey; c) advection of $CH_4$-enriched water from an upstream bubble plume source, not detected by the echosounder. In our case, the temperature- and salinity anomaly, which coincides with the increased
$CH_4$, reveals mixing of AW with colder and fresher water (Fig. 6). Because mixing lines drawn in the TS diagrams (Fig. 2b and 2c) point towards PW rather than a pure fresh water source, our data supports hypothesis c, namely that AW mixed with PW was transported downslope and downstream with the WSC, and was enriched in $CH_4$ while passing over a bubble plume before reaching the location of the measurement.

The 2D model relies on known bubble plume locations, and the difference between measured and modelled $CH_4$ is obvious
along line 3 from 10:30 to 10:50 as seen in Fig. 6. The $CH_4$ signal from high-resolution data, not thoroughly resolved by the model, underscores that mapping and modelling based on echosounder data are not enough for a correct quantitative estimate of the $CH_4$ inventory. The 2D model required a high diffusion coefficient in order to reproduce the variability of measurements, which is supported by high turbulence in the area, caused by the strong currents. Downstream tailing of $CH_4$ concentrations seen in the 2D model was not observed with the MILS. In fact, MILS data reveal equal distribution of down-and upstream
concentration gradients. We explain the discrepancy by the fact that the 2D model does not resolve eddies and the $CH_4$ source is placed in discrete cells, following a theoretical straight bubble line, and not accounting for diffusion along the bubble paths.





The relatively high midwater (120–260 mbsl) $CH_4$ concentrations revealed by the vertical MILS cast 1616 was only partly

observed in the discrete sampling and was not inferred from echosounding. We suggest that this discrepancy is attributed to

seepages at the corresponding depth interval, not previously mapped. The closest known seepages are a few km away from the

location, at the shallow shelf (50–150 mbsl) and at the shelf-break (~250 mbsl) (Veloso et al., 2015), but it is doubtful that

water masses from these locations can reach the surveyed area, as the WSC is persistently northbound. Unless horizontal

eddies transport $CH_4$ from the shelf-break to this area, this result indicates the existence of undiscovered $CH_4$ bubble plumes

further south, at the depth of the observed anomaly.

The high-resolution data from the MILS results in a significantly lower $CH_4$ inventory than the one obtained from discrete

sampling (47 vs 77 nmol $l^{-1}$) due to the heterogeneous distribution of dissolved $CH_4$. The choice of discrete sample locations

can significantly affect the resulting average concentration. The average $CH_4$ concentration (93 nmol $l^{-1}$) estimated by Graves

et al. (2015) from a box with dimensions $\Delta x = 1m$, $\Delta y = 50$ m, $\Delta z = 75$ m, obtained from a DS transect across the slope, was

substantially higher than our box estimates of 20–39 nmol $l^{-1}$. These two results highlight the need for high-resolution sensing

when estimating $CH_4$ inventories and average $CH_4$ concentrations.

**5 Conclusion**

We have presented new methods for understanding the dynamics of $CH_4$ after its release from the seafloor, coupling for the

first time continuous high-resolution measurements from a reliable and fast $CH_4$ sensor (MILS) with dedicated models. The

MILS sensor was successfully deployed as a towed body from a research vessel and provided high-resolution, real-time data

of both vertical and horizontal dissolved $CH_4$ distribution in an area of intense seepage west of Svalbard. For the first time, we

observed a more heterogeneous $CH_4$ distribution than has been previously presumed.

We employed an inverse acoustic model for $CH_4$ seepage mapping and quantification, which provided the basis for a new 2D

model and a new control volume model, which both agreed relatively well with observations. The 2D model did not reproduce

the symmetric gradients observed with the MILS, which suggests a need to improve the model by including turbulent mixing

enhanced by the bubbles streams.

Despite the large spatial and temporal variability of the $CH_4$ concentrations, a comparison between high-resolution (MILS)

and DS data showed good general agreement between the two methods.

Heterogeneous $CH_4$ distribution measured by MILS matched acoustic backscatter, except for an area with high $CH_4$

concentrations without acoustic evidence of $CH_4$ source. Similarly, high midwater $CH_4$ concentration was observed by the

MILS vertical casts with little evidence of a nearby $CH_4$ source, further supporting that high-resolution sensing is an essential

tool for accurate $CH_4$ inventory assessment and that high-resolution sensing can give clues to undetected sources.

$CH_4$ inventories, given by discrete sampling agreed with those from high-resolution measurements, but sparse sampling may

over- or underestimate inventories, which may have repercussions if the acquired data is used for predicting degassing of $CH_4$



to the atmosphere in climate models. The added detail of the fine structure allows for better inventories, elucidates the heterogeneity of the dissolved gas, and provides a better insight to the physical processes that influence the $CH_4$ distribution.

The methods for understanding $CH_4$ seepage presented here shows potential for improved detection and quantification of dissolved gas in oceans and lakes. Applications for high-resolution $CH_4$ sensing with the MILS include environmental and climate studies as well as gas leakage detection desired by fossil fuel industry.

**6 Data availability**

A dataset comprising the MILS sensor data and echosounder files is available from the UiT Open Research Data repository

https://doi.org/10.18710/UWP6LL.

**7 Author contribution**

PJ and JT contributed equally to the work leading to this manuscript, and are listed alphabetically. JM and JC initiated the collaboration and designed the study. JT, RG, PJ, and JM planned and participated in the field campaign. JT and RG systematized and synchronized the high-resolution data. PJ mapped and quantified gas seepage and water currents by collecting

and analysing echosounder data. PJ collected and analysed the CTD data and water samples. RG converted mixing ratios to aqueous concentrations. PJ developed the numerical models. AS analysed the water mass properties. JT and PJ calculated $CH_4$ inventories. All authors contributed to the manuscript.

**8 Acknowledgements**

The research leading to these results has received funding from the European Community's Seventh Framework Programs

ERC-2011-AdG under grant agreement n° 291062 (ERC ICE&LASERS), as well as ERC-2015-PoC under grant agreement n° 713619 (ERC OCEAN-IDs). Additional funding support was provided by SATT Linksium of Grenoble, France (maturation project SubOcean CM2015/07/18). The collaboration between CAGE and IGE was initiated thanks to the European COST Action ES902 PERGAMON. The research is part of the Centre for Arctic Gas Hydrate, Environment and Climate (CAGE) and is supported by the Research Council of Norway through its Centers of Excellence funding scheme grant No. 223259. We

thank the crew on board RV Helmer Hanssen for the assistance during the cruise, and the University of Svalbard for the logistics support.





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

**Table 1: Average concentrations (nmol l⁻¹) calculated with different methods at different altitudes as indicated in the first column**
**(metres above the seabed, masf). [1]Average of the sparse discrete sampling. [2]Average of high-resolution (MILS) measurements. [3]Average of linearly interpolated concentrations based on discrete measurements. [4]Average concentrations from the 2D model, extracted from depths matching the MILS position. [5]Average concentrations within the box (4500 m (L) × 50 m (W) × 75 m (H)) based on the high-resolution measurements or interpolated concentrations along the box, the vertical dissolution-profile, and the mean horizontal concentration gradient across the width of the box. [6]Average of the 2D model best simulation result from 0–75 masf.**
**[7]The CV model yields a box value only.**

| Dataset | Discrete | High-resolution | Box |
| --- | --- | --- | --- |
| MILS ~15 masf | - | 47[2] | 22[5] |
| DS ~5 masf | 104[1] | 108[3] | 39[5] |
| DS ~15 masf | 77[1] | 77[3] | 29[5] |
| DS ~25 masf | 44[1] | 49[3] | 20[5] |
| 2D model (~15[4] and 0–75[6] masf) | - | 60[4] | 22[6] |
| CV model | - | - | 23[7] |