# Peer review of "High-resolution under-water laser spectrometer sensing provides new insights to methane distribution at an Arctic seepage site"

_Ocean Science, 2019_

## Referee Comment (RC1) · Anonymous Referee #1 · 30 May 2019

General Comments: As pointed out by the authors this is a FIRST, and hopefully the instrumentation described will enable a new era of high-quality data to be gathered for ocean and climate studies. The authors can document and quantify both the temporal and spatial heterogeneity of CH4 concentrations in the water column. That such heterogeneity exists is not new, but that it can be quantitatively studied is new. So far technology has limited researchers to either discrete sampling or use of sensors with long response times both making it practically impossible to describe the heterogeneities described in the present study. Coarse data allows for coarse models and budgets. This becomes evident in the data analysis presented. Although the data is high resolution, general applicability of the method for inventory (budgets) studies requires a large

amount of auxiliary data (current, CTD, TS, background/reference measurements). But this is the everyday challenge of the oceanographer (and modeller). The data will allow for substantial discussions within the modelling community. Hopefully, in the future we will see sensors with similar characteristics to that of the "MILS" fitted to groups/swarms of AUV's that can do concurrent sampling and monitoring of larger regions. This could enable true high-resolution characterization of a region of interest and enable high resolution modelling of $CH_4$ dispersion dynamics. Such data will need to be collected in order to be able to use "bottom up" studies to build confidence in "top down" data and models used for inventory monitoring at the ocean and climate scales.

Specific comments: Page 5 L114-117: Where was the pump inlet located? This is not described in the paper nor in Grilli et al. 2018. A schematic is provided of the membrane assembly in SI3 of Grilli et al. 2018.

Page 6 L121-129: Regarding the position correction. A cylinder of height and width of the MILS probe was used. The assembly in Figure 1b show that the CTD, Battery and commercial $CH_4$ sensor is far from symmetric, and the drag of these side mounted addons should probably have been accounted for in the position correction. These addons could also lead to a wobbling and rotation of the assembly. Was this monitored by onboard IMU sensors (inertial measurement unit)?

Section 3.1 Water properties It is not clear from the text that the current information is derived from data obtained simultaneously with the $CH_4$ measurements. This is however stated in Jansson (2019) Figure 8b. When interpreting the inclination of the flairs is flair inclination perpendicular to the ship motion taken into account?

There can of course be unknown sources of the $CH_4$, but there is mention of WSC meandering, and negligible tidal effects. Have typical eddy sizes been characterized? The time between transect lines 1 and 5 are by rough estimation 12 hrs i.e. roughly one tidal period. The whole cruise was two tidal periods. What is the direction of the tidal flow in this region? Both eddy size and tidal currents could result in noticeable

advection over a 12-hr period.

Page 13 L267: with the given speed of the cruise and the response time of the instrument (15 sec), spatial resolution is of the order 10m. However, how does the instrument obtain a measurement? Is it by continuous flow at a given flow rate over the membrane, or does it work in a batch mode with discrete samples passed over the membrane unit?

Page 13 L280: What is the reasoning behind scaling up the flair by 40%? Can the authors justify this quantitatively?

Technical corrections: Page 4 – L62-75 A map/graphic could be included for illustration if authors have access to graphical assistance. Page 4 – L80 and L95-97: purely cosmetic but I like it when lists come in the same order, e.g. temp, salinity and concentration. Page 7 – L150-180: I feel that the presentation in paragraphs 2.5 and 2.6 could benefit from a graphic illustrating the computational domains. I believe that this will aid the reader in understanding and conceptualizing the differences between the two methods better.

Figure 2: Second line: it should read "Gibbs seawater package". In the last line: the mean bubble rise velocity is 23 cm s-1, could you provide the mean bubble size as well?

Figure 7: The figure would be much easier to read if it was in colour.

---

## Referee Comment (RC2) · Anonymous Referee #2 · 5 Jun 2019

General comments The manuscript describes how a new technology/sensor can improve our knowledge on the distribution and the dynamics of CH4 over an Arctic seep area. This technology uses a laser spectrometer and a membrane inlet to extract the gas from the aqueous phase. The manuscript is clearly written, results and discussion are well presented, although a bit confusing when it gets to the description of the models (a schematic/conceptual model would have been appreciated).

Without any doubt, the lack of in situ, high-resolution measurements of methane in marine environments makes difficult to fully understand their role as a source and/or a sink of methane. This is probably for this reason the contribution of the oceans to

the global methane budget has been underestimated. So every effort to develop and test new sensors and technologies must be encouraged. In that regard, the manuscript does represent a significant contribution towards a better comprehension of the marine methane cycle, and therefore, deserves to be published in OS, upon minor revision.

However, I would not say this is a first. Yes it is the first time that this particular technology is deployed in operational conditions – with satisfying results – but this is not the first attempt to get a high-resolution map of CH4 distribution in marine environments. Just to name a few studies on the subject: Sommer et al 2015 (10.1016/j.marpetgeo.2015.08.020), Gentz et al 2014 (10.4319/lom.2012.10.317), Wankel et al 2009 (10.1016/j.dsr2.2010.05.009)... Perhaps, this new sensor has better performance in terms of detection limits and response time, but it's very hard to find them in the manuscript. How does the MILS compare to them?

The development of CH4 sensors has been the holy grail for decades now, and a few technologies emerged from this effort. Each of them were considered as the new solution but I think the main mistake is to believe that one instrument can address the full range of concentrations encountered in the ocean – from 0.1 nM to several mM. This is of course not possible and the instrument must be adapted to the scientific question. In that regard, the MILS seems to be very well adapted to the environmental conditions in which it was deployed. Can the MILS be deployed in oligotrophic waters, i.e. at very low concentrations? And can it measure very high concentrations like in the Black Sea or in the Baltic Sea?

One big question at the moment is the role of phytoplankton blooms on the emission of methane to the atmosphere. There are many areas in the open ocean that are characterised by methane anomalies in the upper layer (i.e. the ocean methane paradox). Concentrations are not necessarily very high (up to 5 nM) but enough to oversaturate the upper layer, and therefore create a positive flux to the atmosphere. Is the MILS able to measure concentrations in this range? I think the effort must be now pointed to low concentration measurements. Anyhow, if one can adapt this instrument to lower

concentrations, and then if it can be deployed on AUVs (or any other autonomous platforms), then we will definitely advance the knowledge on the marine CH4 budget. The ideal would be to use this kind of instruments for process studies, i.e. in situ measurements of production/ consumption rates, which will further advance the comprehension of the biogeochemical cycle of methane.

Specific comments Line 28: I would rephrase 'contributing to minimum oxygen zone formation, and possibly to ocean acidification, as a result of the oxidation of methane'. This last point is still under debate as it is impossible to evaluate precisely the contribution of methane oxidation to the production of CO2 (again because of the lack of in situ data). And yet, the dynamics of these 2 gases are very different.

Lines 40 to 49. I would moderate the discussion here. I think we should view echosounding as a complementary technique to dissolved gas measurements. The big advantage of the echosounding technique is to locate seeping areas while measuring only dissolved methane cannot help deciphering the sources. As for example, in the Black Sea, concentrations are so high that it is impossible to detect the seeping areas other than using echosounding. One advantage I can see is to evaluate the dynamics of bubble dissolution in the water column as gas bubbles are a mean of transfer of methane from the bottom to the surface.

Line 53 I would not put in situ mass spectrometry away so quickly. It is commonly used in deep sea studies, especially in hydrothermal environments. Check Boulart et al. 2017, G3. It may have a slower response time but its main advantage is the ability to detect and measure several analytes in the same time.

Line 101 What is the autonomy of the MILS? What is the power consumption?

Line 101 So the MILS uses exactly the same sample introduction system as in situ mass spectrometers. I guess this is the same kind of PDMS membranes? As the authors wrote, membranes are sensitive to fluctuating water flow. I would add 'pressure of deployment' as well. Membrane's permeability is not the same when deployed at the

surface and at 100m depth. How is the pressure effect calibrated? A table comparing MILS' performance with other instruments would be useful here, so the reader can see the advantages of using MILS rather than an ISMS or something else.

Line 114 What do the authors mean by 'careful positioning of the SBE5T'? How do they minimize the pressure change? Is the pump very close to the membrane inlet?

Lines 185-195 Was it obtained with the SBE or with the Anderaa?

Section 3.2 Why do the author use 'm above seafloor' as the vertical scale for their casts? This is unusual and can be confusing for the reader. Please use 'm below sea level' for all vertical casts. When is the pump started during vertical or horizontal casts? I guess it is a continuous flow?

Lines 349-356 The authors do not mention the possibility of methanotrophy (microbial oxidation), which is the main control of the vertical distribution of methane in the water column. They can refer to the studies in the Black Sea where concentrations close to the seafloor is up to 12 $\mu$mol/l. See Schmale et al 2011, BGS.

---

## Referee Comment (RC3) · Anonymous Referee #3 · 11 Jun 2019

[referee-annotated manuscript omitted]

---

## Author Comment (AC1) · 13 Jul 2019

Authors replies to the interactive comments of anonymous referee #1 (30 May 2019) on "High-resolution under-water laser spectrometer sensing provides new insights to methane distribution at an Arctic seepage site" by Pär Jansson et al.

**RC:** denotes referee's comments

**AR:** denotes authors' reply

**MC:** denotes manuscript changes

**RC:** General Comments: As pointed out by the authors this is a FIRST, and hopefully the instrumentation described will enable a new era of high-quality data to be gathered for ocean and climate studies. The authors can document and quantify both the temporal and spatial heterogeneity of CH4 concentrations in the water column. That such heterogeneity exists is not new, but that it can be quantitatively studied is new. So far technology has limited researchers to either discrete sampling or use of sensors with long response times both making it practically impossible to describe the heterogeneities described in the present study. Coarse data allows for coarse models and budgets. This becomes evident in the data analysis presented. Although the data is high resolution, general applicability of the method for inventory (budgets) studies requires a large amount of auxiliary data (current, CTD, TS, background/reference measurements). But this is the everyday challenge of the oceanographer (and modeller). The data will allow for substantial discussions within the modelling community. Hopefully, in the future we will see sensors with similar characteristics to that of the "MILS" fitted to groups/swarms of AUV's that can do concurrent sampling and monitoring of larger regions. This could enable true high-resolution characterization of a region of interest and enable high resolution modelling of CH4 dispersion dynamics. Such data will need to be collected in order to be able to use "bottom up" studies to build confidence in "top down" data and models used for inventory monitoring at the ocean and climate scales.

**AR:** Thank you for taking the time and effort to read and comment on our manuscript. You acknowledge the need for this kind of research, and recognize the hardship of acquiring useful data, even with the new advanced technology. We appreciate your comments, which we believe has improved the manuscript. We agree that this type of high-resolution measurements is the beginning of a new era of Oceanographic surveys, and that more data, both in time and space, is needed for a broader understanding.

**RC:** Specific comments: Page 5 L114-117: Where was the pump inlet located? This is not described in the paper nor in Grilli et al. 2018. A schematic is provided of the membrane assembly in SI3 of Grilli et al. 2018. Page 6 L121-129:

**AR:** The water circulation pump (Seabird SBE 5T) was located at the bottom of the MILS instrument approx. 25 cm away from the membrane assembly block. Short sections of ½" hose and a T-piece were used to connect the pump outlet to the membrane block inlets. The pump inlet was shielded with a cover and a fine mess to avoid ingress of particles and/or bubbles.

**MC:** This information was added to the text in lines 121–123.

**RC:** Regarding the position correction. A cylinder of height and width of the MILS probe was used. The assembly in Figure 1b show that the CTD, Battery and commercial CH4 sensor is far from symmetric, and the drag of these side mounted addons should probably have been accounted for in the position correction. These addons could also lead to a wobbling and rotation of the assembly. Was this monitored by onboard IMU sensors (inertial measurement unit)?

**AR:** For the position correction, a cylinder shape was used with a height and diameter equivalent to the displacement/buoyancy of the total assembly of instruments (i.e. not just the height and width of the MILS). This ensured that the simulated buoyancy of the total assembly was as close to reality as possible to allow for the (unknown) drag coefficient to be determined by making the simulation match with all the other known parameters such as instrument depth, cable length, ship speed, ship direction, and currents. It is unknown how stable the instrument assembly was while being towed, but wobble and/or rotation would have no significant effect on the measurements. No IMU sensors were used to monitor the movement of the assembly during profiling.

**MC:** This information has now been added to the discussion in lines 364–365

**RC:** Section 3.1 Water properties It is not clear from the text that the current information is derived from data obtained simultaneously with the CH4 measurements. This is however stated in Jansson (2019) Figure 8b.

**AR:** We agree that this should be more clearly stated. We added a note on that in the manuscript

**MC:** lines 204–206 state the above.

**RC:** When interpreting the inclination of the flairs is flair inclination perpendicular to the ship motion taken into account?

**AR:** The split-beam echosounder (Simrad EK60) resolves the location of scattering objects in 3 dimensions, but the echosounder swath width (~7°) will set a limit to the positions of the scattering objects. Particularly, in the direction perpendicular to the ships' movement, the bubbles may easily escape the beam if the current carries them across the ship trajectory. During our survey, the heading of the ship is biased towards the N/S and S/N direction, and it is therefore possible for flares to extend more in that direction. However, careful investigation of the flare data shows that the flares detected during cross-slope sailing have very small east-west components even though they could potentially extend across-slope within the echosounder beam. We are therefore convinced that our flare-inferred currents represent real currents. Furthermore, ocean currents generally flow along isobaths, and the streamlines determined by potential vorticity conservation, follow the isobaths closely in this area Nilsen et al. (2016).

**RC:** There can of course be unknown sources of the CH4, but there is mention of WSC meandering, and negligible tidal effects. Have typical eddy sizes been characterized? The time between transect lines 1 and 5 are by rough estimation 12 hrs i.e. roughly one tidal period. The whole cruise was two tidal periods. What is the direction of the tidal flow in this region? Both eddy size and tidal currents could result in noticeable advection over a 12-hr period.

**AR:** Eddies are difficult to observe with sparse observations, but high resolution modelling suggests that mesoscale (a few km) eddies are important for transport of water properties across the slope in the study area. Mesoscale and smaller eddies form on each side of the WSC core, which also meanders off- and onshore of our study site (Hattermann et al., 2016, supplementary information). This process obviously affects also methane concentrations, which could appear high or low without other obvious explanations. We do not discard the possibility that eddies transport $CH_4$ enriched water in ways that we cannot predict without perfect knowledge of the velocity field. We simply put forward the possibility that unknown sources could be tracked with the new instrument. The $CH_4$ enriched water that we observe in the northern part of line 3, not explained by acoustically observed flares, was accompanied with a TS anomaly. This suggest intrusion of a different water mass, but not

all of the intrusion was enriched with $CH_4$. Possibly, this is eddy induced, or it could be a result of bottom Ekman transport.

**MC:** We added the possibility for eddies in the discussion (line 385).

**AR:** The survey lasted for three days (October 21$^{st}$ – 23$^{rd}$). The probe was deployed each morning around 10 AM, and was measuring continuously for 4, 9, and 10 hours respectively. The tidally driven currents in the area range between -1 and 1 cm s$^{-1}$ in both the east and north directions. The probe was deployed at the approximate same tidal state and the modelled tides during the deployments were 0–1 vs -0.5–0.5 cm s$^{-1}$ in the N and E direction respectively.

**RC:** Page 13 L267: with the given speed of the cruise and the response time of the instrument (15 sec), spatial resolution is of the order 10m. However, how does the instrument obtain a measurement? Is it by continuous flow at a given flow rate over the membrane, or does it work in a batch mode with discrete samples passed over the membrane unit?

**AR** Both the water flow over the membranes and the gas flows inside the instrument are continuous and constant during the cast/deployment.

**RC:** Page 13 L280: What is the reasoning behind scaling up the flair by 40%? Can the authors justify this quantitatively?

**AR:** The 40% upscaling is based on the "dissolution function" or "non-dimensional source-function" (sect 2.6), which shows that a large portion of the initial $CH_4$ is already lost from the bubbles when we observe them with the echosounder in the layer 5 – 10 masf.

**MC:** The upscaling due to dissolution is now better explained in line 292.

**RC:** Technical corrections: Page 4 – L62-75 A map/graphic could be included for illustration if authors have access to graphical assistance.

**AR:** An illustration with currents carrying the different water masses would be nice, but it is outside the scope of this paper to produce an infographic on water mass movements. The physical oceanography is well documented in the referenced papers and we do not wish to review them extensively in our manuscript. To partly meet your suggestion, we added the main controlling ocean currents in figure 1a (inset map). Water mass classifications are found in the TS-diagram in Figure 2b and 2c.

**MC:** Figure 1 was updated and now indicates the dominating currents.

**RC:** Page 4 – L80 and L95-97: purely cosmetic but I like it when lists come in the same order, e.g. temp, salinity and concentration.

**AR:** We agree.

**MC:** Order of parameters changed in line 102.

**RC:** Page 7 – L150-180: I feel that the presentation in paragraphs 2.5 and 2.6 could benefit from a graphic illustrating the computational domains. I believe that this will aid the reader in understanding and conceptualizing the differences between the two methods better.

**AR** We posted a supplementary containing a schematic showing the control volume and the 2D model, which indicates the included processes for easier understanding.

**MC:** Fig. SI 1 was added in the supplementary document.

**RC:** Figure 2: Second line: it should read "Gibbs seawater package". In the last line: the mean bubble rise velocity is 23 cm s-1, could you provide the mean bubble size as well?

**AR:** Thanks for noticing that. We corrected the caption for figure 2. We added the bubble size distribution.

**MC:** Manusript changed accordingly. Bubble size distribution in line 159.

**RC:** Figure 7: The figure would be much easier to read if it was in colour.

**AR:** Ok.

**MC:** Figure 7 and its caption has been updated accordingly.

Hattermann, T., Isachsen, P. E., Appen, W. J., Albretsen, J. & Sundfjord, A. 2016. Eddy-driven recirculation of Atlantic Water in Fram Strait. *Geophysical Research Letters*, 43(7), pp. 3406-3414.

Nilsen, F., Skogseth, R., Vaardal-Lunde, J. & Inall, M. 2016. A Simple Shelf Circulation Model: Intrusion of Atlantic Water on the West Spitsbergen Shelf. *Journal of Physical Oceanography*, 46(4), pp. 1209-1230. doi: 10.1175/jpo-d-15-0058.1.

---

## Author Comment (AC3) · 13 Jul 2019

Authors replies to the interactive comments of anonymous referee #3 (11 June 2019) on "High-resolution under-water laser spectrometer sensing provides new insights to methane distribution at an Arctic seepage site" by Pär Jansson et al.
* * *
**RC:** denotes referee's comments

**AR:** denotes authors' reply

**MC:** denotes manuscript changes
* * *
**RC:** The manuscript by Pär Jansson et al "High-resolution under-water laser spectrometer sensing provides new insights to methane distribution at an Arctic seepage site" describes the application of a new methane sensor to methane seeps off Svalbard. As the sensor measures methane in situ with a high temporal resolution a very accurate methane inventory of this probably highly variable area is given. Overall the Ms is well written and straightforward. However, the figures contain too much information, which is either not well explained or not necessary for the specific message, and thus are sometimes rather confusing. For a "non-modeller" I found it sometimes difficult to follow the outline of the applied models. In the discussion, both the technical and the scientific aspects should be discussed, But both are rather short. I would be interested in the comparison with the other commercial methane sensor which was attached to the device. . ... Also an estimation on which temporal resolution is really necessary to dissolve the methane distribution would be appreciated, and what influence has the towing speed on the pattern?? More detailed comments can be found in the attached pdf-file

**AR:** Thank you for taking the time and effort to read and comment on our manuscript. You acknowledge the need for this kind of research, and recognize the hardship of acquiring useful data, even with the new advanced technology. We appreciate your comments, which we believe improved the manuscript.

Regarding your comment that some of the figures contain too much information, Figure 3 has been split into two parts and the information about concentration gradients have been moved to a supplementary section. We also improved the resolution of figure 4. We prefer to keep the "MILS all" in the figure, since it shows the general vertical distribution, which is not shown anywhere else in the manuscript. We put the inset maps in the figure to visualise the origin of the different data points so it would be easier to see the spatial separation between them.

We now supply a visualisation of the model domains in the supplementary information for a better understanding of the control volume and 2D model, as also requested by two other referees.

The technical details of the MILS has been largely omitted in this manuscript, because the instrument has already been thoroughly described in a previous publication (Grilli et al., 2018) . Here, we evaluate its functionality in this particular environment, with a focus is on what we can learn from the high-resolution measurements. This is why the technical discussion is short.

We agree that a comparison with the reference sensor would be interesting. However, for this publication, we decided to refrain from a direct sensor comparison, as we would rather leave it to a non-biased future publication to compare the MILS, the reference sensor and other $CH_4$ sensors in a more technical publication.

Regarding the resolution, towing speed and the sampling frequency, we believe that we have good enough resolution, since the MILS picks up the concentration gradients along the ship track. See lines 278–280 and the new Fig. SI 2 in the supplementary information. If we want better resolution in

three dimensions, we would need denser surveys, and so it will always be a trade-off between costs and data resolution.

*Hereafter is a list of comments and suggestions from the referee, which was posted as pop-up notes in a supplement to the comments. Care has been taken to include all comments and suggestions, and answers given to the best of our ability.*

**RC:** Line 99: At which speed was the ship moving ??

**AC:** The ship's position was logged continuously and can be found in the file in the data repository. During Line 3, the average speed was 0.79 m/s (1.5 knots) with a standard deviation of 0.065 m/s (0.13 knots).

**MC:** We added a note on the speed on line 105.

**RC:** Line 101: Can you give an estimate on the accuracy of this distance ??

**AC:** We added this information on lines 254–255.

**RC:** Line 151: What is meant with control volume ??

**AR:** In engineering literature, a control volume is a region fixed in space and its surfaces are called control surfaces. (e.g. Kundu et al., 2008).

**MC:** We now clarify it on line 164

**RC:** Line 186: In October 2015, ......

**AR:** We rephrased this sentence

**MC:** Line 200 was changed accordingly.

**RC:** Line 186: I find the two "depth" or "height" definitions confusing, I suggest to use only one of them…

**AC:** We changed the phrase so it is easier to read, and are now avoiding usage of two different abbreviations.

**MC:** Lines 200–201 were changed accordingly

**RC:** Line 222–225: That is too much information in one figure…I suggest to shift the inlet and additional infos on line 3 into an extra figure… The gradient story is not mentioned in the text, thus if rather confusing here…

**AR:** We followed your advice and split this information into two figures. Figure 3 now focuses only on the concentrations along the five main trajectories. The caption has been truncated accordingly. A new figure, visualising the gradients, is included in the supplementary document (Figure SI 2). The gradients are discussed on lines 272–283.

**MC:** New figure 3. Caption of figure 3 truncated. New figure in supplementary document (Fig. SI 2).

**RC:** Line 226–229 : This technical information should go either M&M or

**AR:** We now describe this in the methods section, and mention the results in the appropriate section.

**MC:** Lines 139–141 and lines 236–238

**RC:** Line 230: Again, there is too much information in the figure 4, which is then not mentioned in the text... please refrain to the important facts. If you only want to compare the vertical casts of CTD 616, 618,619 than all other informations are not needed and are more confusing....

**AR:** We believe that visualising the different measurements together with their spatial separation is key to understanding the heterogeneity of the $CH_4$ distribution. It may take some effort to appreciate this figure, but we think it is valuable to show the spatial separation together with the concentration data, in order to realise the distribution of dissolved $CH_4$.

**RC:** Line 247: But there are also areas with strong bubble streams but with low methane concentrations ?? For example at the very left side of the figure ??

**AR:** From our experience with echosounder data, no bubble streams are visible in the echogram on the left side of the figure. Conversely, there are methane peaks without visible flares, which we discuss at some length on lines 374–385. On the right hand side of the figure, there are flares without increased concentrations. This may be due to the fact, that the echosounder swath width is ~40 m at the seabed, while the MILS measures locally. Therefore, it is possible that we passed nearby a flare, which was acoustically identified but that we were too far away to see the $CH_4$ plume in the MILS data.

**RC:** Line 249: Again too much information here: what for are the upper inserts needed? And MILS all ?? What about the other DS from the casts along line 3, 1623, 1621, ff ??

**AR:** See our comment above about line 230. Comparison between the discrete samples from CTDs, and MILS data from line 3 data can be seen in Figure 7, which has been improved also after the request from another referee.

**MC:** Figure 7 has been improved

**RC:** Line 250: remove from figure and legend

**AR:** See our answer regarding line 230 and 249. We believe that this visualisation helps to understand the vertical distribution.

**RC:** Line 255–259: remove from figure and legend, I think it is sufficient to mention that the CTD casts were xx m away.

**AR:** See our answer regarding line 230 above.

**RC:** Line 262: The blue line is hardly visible, but propably also not necessary as already shown in figure 1.

**AR:** The blue line does not stand out very well in this document, but it looks good in the original figure. We think it will be clear in the final version without any changes. The instrument position is shown in Figure 1, but the depth is not indicated. The depth is shown in Figure 3, but has no reference. In this figure, it is presented to scale with the echosounder data, which we believe is important for the interpretation.

**RC:** Line 265: I do not understand what is meant with upstream and downstream gradient, and thus also can not follow your conclusions…

**AC:** We have removed the gradients from figure 3 and moved this information to the supplementary information. In the new figure (SI 2), the upstream/ downstream gradients are explained and visualised. Thanks for directing our attention to this. It was not very clear earlier.

**MC:** New Fig. 3 and new Fig. SI 2.

**RC:** Line 268–276:  ???? I can not follow here and I am not sure if this information on the instrument characteristics is necessary here, as this should have been done in the previous publication and not its application now…

**AR:** Here we argue that the instrument has good enough resolution for this particular environment. It is not about the technical aspects of the instrument itself, but that we managed to resolve the $CH_4$ distribution by towing the probe with the right settings at the right speed, so it could pick up the heterogeneity.

**RC:** Line 292: could you indicate the stream / current in the figure ?

**AR:** Yes, of course.

**MC:** We added an arrow in this figure and in Figure SI 2.

**RC:** Line 297: But also the water depth of the instrument was more stable in this area, compared to the fluctuation before and after…

**AR:** After double-checking the data, we found that the relative standard deviations of the probe depth, salinity, and temperature is lower by factors of 3, 10, and 58, when compared to the upstream (later in time) data. This is consistent with the notion that the "flat profiles" between 10:50 and 11:15 are caused by enhanced turbulent mixing due to bubble streams. The standard deviation of the probe depth dropped by a factor of 3 which is not enough to explain the larger drops in salinity and temperature standard deviations. It is normal that temperature diffuses faster than salinity (see textbooks on ocean turbulence), so the fact that the temperature profile is flatter than the salinity profile has a reasonable explanation. We re-phrased the sentence to describe this feature more accurately. Thanks for noticing that.

**MC:** Lines 305 – 309

**RC:** Line 302: but is it a good match ?? the methane peaks on the left side are not resolved in the modell and at the right side the model seems to be shifted…

**AR:** We are not arguing that this is a good match. It is simply the best of the performed simulations, which are solely based on flare observations and the assumptions of a homogenous, steady water current along the domain, homogenous and constant diffusion etc. We do not expect a perfect match from such a simple model, but find it striking that the model does so well with so little information. The lack of sources for the downstream (left) peaks are mentioned in the discussion. We do not have an immediate explanation for the apparent shift on the right hand side of the figure. It could be due to undiscovered sources, imperfect time lag correction of the instrument data, wrong assumption of homogenous water current or it can be explained by the relatively large swath of the echosounder while the MILS measure locally (see our reply to the RC comment about line 247).

**RC:** Line 326: I find it difficult to understand how you calculated the average concentration of the specific area and being a non-modeller, when you compare a average shouldn't there be a standard deviation ?? To judge if 47 is about the same as 77 nM. ??

**AR:** The model domain is now visualised in the supplementary information and the improved Figure 7 shows the discrete and high-resolution data, which underlies the average calculations. The caption for Table 1 explains how the data was averaged. The point is that high-resolution data makes a better estimate for a $CH_4$ inventory, while sparse sampling can easily over- or underestimate the inventory.

Standard deviation does not make sense here, but one should keep in mind the uncertainties of the methods (4% for the discrete samples and 12% for the MILS data). The model builds on "flare quantification" with uncertainties related to bubble sizes and rising speeds, discussed at length in Veloso et al. (2015). The model has a correlation with the high-resolution data of 0.68, so should be evaluated with care.

**RC:** Line 353: but still below the pycnocline ... aha.. this was well below the pyccnocline, thus background levels of methane were reached below the pycnocline, which there fore could not act as barrier... Maybe you should re-phrase your argumentation here...

**AR:** That would be a way of saying the same thing, but it does not help to understand the mechanism. We believe it is appropriate to give a plausible explanation, rather than just stating the obvious fact that vertical transport is inefficient. Here we explain why it is not necessary to have a pycnocline to impede vertical transport of solvents (in this case dissolved $CH_4$). A small continuous stratification is enough. The argument that wintertime stratification-breakdown can cause sudden emissions of $CH_4$ to the atmosphere still stands.

Grilli, R., Triest, J., Chappellaz, J., Calzas, M., Desbois, T., Jansson, P., Guillerm, C., Ferré, B., Lechevallier, L., Ledoux, V. & Romanini, D. 2018. Sub-Ocean: Subsea Dissolved Methane Measurements Using an Embedded Laser Spectrometer Technology. *Environmental Science & Technology*, 52(18), pp. 10543-10551. doi: 10.1021/acs.est.7b06171.
Kundu, P. K., Cohen, I. M. & Dowling, D. 2008. Fluid Mechanics 4th. Elsevier.
Veloso, M., Greinert, J., Mienert, J. & De Batist, M. 2015. A new methodology for quantifying bubble flow rates in deep water using splitbeam echosounders: Examples from the Arctic offshore NW-Svalbard. *Limnology and Oceanography: Methods*, 13(6), pp. 267-287. doi: 10.1002/lom3.10024.

---

## Author Comment (AC2)

Authors replies to the interactive comments of anonymous referee #2 (5 June 2019) on "Highresolution under-water laser spectrometer sensing provides new insights to methane distribution at an Arctic seepage site" by Pär Jansson et al.

**RC: denotes referee's comments**

**AR: denotes authors' reply**

**MC: denotes manuscript changes**

**RC**: General comments The manuscript describes how a new technology/sensor can improve our knowledge on the distribution and the dynamics of CH4 over an Arctic seep area. This technology uses a laser spectrometer and a membrane inlet to extract the gas from the aqueous phase. The manuscript is clearly written, results and discussion are well presented, although a bit confusing when it gets to the description of the models (a schematic/conceptual model would have been appreciated). Without any doubt, the lack of in situ, high-resolution measurements of methane in marine environments makes difficult to fully understand their role as a source and/or a sink of methane. This is probably for this reason the contribution of the oceans to the global methane budget has been underestimated. So every effort to develop and test new sensors and technologies must be encouraged. In that regard, the manuscript does represent a significant contribution towards a better comprehension of the marine methane cycle, and therefore, deserves to be published in OS, upon minor revision.

However, I would not say this is a first. Yes it is the first time that this particular technology is deployed in operational conditions – with satisfying results – but this is not the first attempt to get a high-resolution map of CH4 distribution in marine environments. Just to name a few studies on the subject: Sommer et al 2015 (10.1016/j.marpetgeo.2015.08.020), Gentz et al 2014 (10.4319/lom.2012.10.317), Wankel et al 2009 (10.1016/j.dsr2.2010.05.009)... Perhaps, this new sensor has better performance in terms of detection limits and response time, but it's very hard to find them in the manuscript. How does the MILS compare to them?

The development of CH4 sensors has been the holy grail for decades now, and a few technologies emerged from this effort. Each of them were considered as the new solution but I think the main mistake is to believe that one instrument can address the full range of concentrations encountered in the ocean – from 0.1 nM to several mM. This is of course not possible and the instrument must be adapted to the scientific question. In that regard, the MILS seems to be very well adapted to the environmental conditions in which it was deployed. Can the MILS be deployed in oligotrophic waters, i.e. at very low concentrations? And can it measure very high concentrations like in the Black Sea or in the Baltic Sea? One big question at the moment is the role of phytoplankton blooms on the emission of methane to the atmosphere. There are many areas in the open ocean that are characterised by methane anomalies in the upper layer (i.e. the ocean methane paradox). Concentrations are not necessarily very high (up to 5 nM) but enough to oversaturate the upper layer, and therefore create a positive flux to the atmosphere. Is the MILS able to measure concentrations in this range? I think the effort must be now pointed to low concentration measurements. Anyhow, if one can adapt this instrument to lower concentrations, and then if it can be deployed on AUVs (or any other autonomous platforms), then we will definitely advance the knowledge on the marine CH4 budget. The ideal would be to use this kind of instruments for process studies, i.e. in situ measurements of production/ consumption rates, which will further advance the comprehension of the biogeochemical cycle of methane.

**AR:** Thank you for taking the time and effort to read and comment on our study. You have acknowledged the importance of this type of investigations. We feel confident that we will see more high-resolution surveys of the same type in the future. In your general comments, you specifically asked for a graphic describing the numerical models, which also reviewer #1 asked for. We added an illustration along with a caption as a part of a new supplementary document.

Regarding the instrument capability and how it compares to other instruments, we refer to the study of Grilli et al. (2018). We would like to avoid an explicit comparison of the MILS to other instruments in this study, and leave that to an impartial instrument comparison study. On page 3, we already mentioned the work of Gentz et al. (2014). Additionally, we now mention the work of Sommer et al. (2015), Wankel et al. (2010), and Boulart et al. (2017) in lines 53–57.

The instrument has a specific range of concentrations as you mention, but for instance, the optical spectrometer can be differently tuned or even replaced to improve its sensitivity or to sample more CH4 enriched waters. The SubOcean (which we call MILS in our study) was deployed in March 2018 at Lake Kivu, measuring up to 3 mM of CH4. The report from the Lake Kivu campaign is found here: https://www.dora.lib4ri.ch/eawag/islandora/object/eawag%3A18541/datastream/PDF/Schmid-2019-Intercalibration campaign for gas concentration-%28published\_version%29.pdf. We believe the MILS would be an excellent tool for evaluating CH4 related water column processes. Grilli et al. (2018) reported a sensitivity of ±25 ppbv in air, translating into ±0.03 nmol L–1 at 20 °C and a salinity of 38, which is low enough for investigations of atmospheric exchange and CH4 production/ consumption rates.

**MC:** We added a graphic (Fig. SI 1) describing the numerical models in the supplementary document. **MC:** The works of Sommer et al. (2015), Wankel et al. (2010), and Boulart et al. (2017) are now mentioned in lines 53-57

**MC:** In lines 408–411, we added a note on the suitability of the MILS for detailed charting of water column processes and ocean-atmosphere interaction.

**RC:** Specific comments Line 28: I would rephrase 'contributing to minimum oxygen zone formation, and possibly to ocean acidification, as a result of the oxidation of methane'. This last point is still under debate as it is impossible to evaluate precisely the contribution of methane oxidation to the production of CO2 (again because of the lack of in situ data). And yet, the dynamics of these 2 gases are very different.

**AR:** To our knowledge, the effect of CH4 oxidation on ocean acidification is today still unknown, and has so far only been modelled. We have rephrased this sentence.

MC: Rephrased sentence in line 27–29.

**RC:** Lines 40 to 49. I would moderate the discussion here. I think we should view echosounding as a complementary technique to dissolved gas measurements. The big advantage of the echosounding technique is to locate seeping areas while measuring only dissolved methane cannot help deciphering the sources. As for example, in the Black Sea, concentrations are so high that it is impossible to detect the seeping areas other than using echosounding. One advantage I can see is to evaluate the dynamics of bubble dissolution in the water column as gas bubbles are a mean of transfer of methane from the bottom to the surface.

**AR:** Clearly, the methods described have their own advantages, and one does not exclude the other. We have edited this section and phrased it differently in order to give a more nuanced picture.

MC: Rephrased sentences in lines 40 - 50

**RC:** Line 53 I would not put in situ mass spectrometry away so quickly. It is commonly used in deep sea studies, especially in hydrothermal environments. Check Boulart et al. 2017, G3. It may have a slower response time but its main advantage is the ability to detect and measure several analytes in the same time.

**AR:** The MILS is by no means the only solution to in situ measurements of CH4, and mass spectrometers has the advantage of measuring different dissolved gas species simultaneously. We now mention the Boulart et al. (2017) survey in the text.

MC: The work of Boulart et al. (2017) is mentioned in lines 57-60

RC: Line 101 What is the autonomy of the MILS? What is the power consumption?

AR: 12 h autonomy at 50W.

**MC**: We now mention the autonomy in lines 99–100.

**RC:** Line 101 So the MILS uses exactly the same sample introduction system as in situ mass spectrometers. I guess this is the same kind of PDMS membranes? As the authors wrote, membranes are sensitive to fluctuating water flow. I would add 'pressure of deployment' as well. Membrane's permeability is not the same when deployed at the surface and at 100m depth. How is the pressure effect calibrated? A table comparing MILS' performance with other instruments would be useful here, so the reader can see the advantages of using MILS rather than an ISMS or something else.

**AR:** Composite PDMS membranes were used. PDMS permeability is not significantly affected by the water pressure at these depths. (Robb, 1968)

**RC:** Line 114 What do the authors mean by 'careful positioning of the SBE5T'? How do they minimize the pressure change? Is the very close to the membrane inlet?

**AR:** The pump was positioned about 25cm away from the membrane inlets and connected with short  $\frac{1}{2}$ " hose sections and a T piece. By shielding the inlet and outlets and mounting them at the same height with an open flow path pressure changes due to movement through the water column were minimised.

MC: We rephrased the sentences in lines 121–123.

RC: Lines 185-195 Was it obtained with the SBE or with the Anderaa?

**AR:** The vertical casts (Figure 2a) were obtained with the Seabird SBE CTD and the TS diagrams (Figure 2b, c) were obtained from the towed Anderaa CTD.

MC: This is now mentioned in the text (lines 200 and 208).

**RC:** Section 3.2 Why do the author use 'm above seafloor' as the vertical scale for their casts? This is unusual and can be confusing for the reader. Please use 'm below sea level' for all vertical casts.

**AR:** We agree that it may be a bit unusual to use 'm above seafloor' for the vertical scale, but we are specifically investigating seepage from the seafloor and found it natural to describe the flow from its source. This approach enabled us to evaluate the distribution resulting from seepage.

RC: When is the pump started during vertical or horizontal casts? I guess it is a continuous flow?

**AR:** The water pump (SBE5T) was started at the surface, and ran continuously for the duration of each deployment.

**RC:** Lines 349-356 The authors do not mention the possibility of methanotrophy (microbial oxidation), which is the main control of the vertical distribution of methane in the water column. They can refer to the studies in the Black Sea where concentrations close to the seafloor is up to 12  $\mu$ mol/l. See Schmale et al 2011, BGS.

**AR:** Methanotrophic oxidation is an important sink on a larger scales and longer time scales, but is locally insignificant at sites with intense CH4 bubble seepage and high water through-flow and therefore short residence times (Jansson et al., 2019).

- Boulart, C., Briais, A., Chavagnac, V., Révillon, S., Ceuleneer, G., Donval, J.-P., Guyader, V., Barrere, F., Ferreira, N., Hanan, B., Hémond, C., Macleod, S., Maia, M., Maillard, A., Merkuryev, S., Park, S.-H., Ruellan, E., Schohn, A., Watson, S. & Yang, Y.-S. 2017. Contrasted hydrothermal activity along the South-East Indian Ridge (130°E–140°E): From crustal to ultramafic circulation. *Geochemistry, Geophysics, Geosystems*, 18(7), pp. 2446-2458. doi: 10.1002/2016GC006683.
- Gentz, T., Damm, E., Schneider von Deimling, J., Mau, S., McGinnis, D. F. & Schlüter, M. 2014. A water column study of methane around gas flares located at the West Spitsbergen continental margin. *Continental Shelf Research*, 72, pp. 107-118. doi: 10.1016/j.csr.2013.07.013.
- Grilli, R., Triest, J., Chappellaz, J., Calzas, M., Desbois, T., Jansson, P., Guillerm, C., Ferré, B., Lechevallier, L., Ledoux, V. & Romanini, D. 2018. Sub-Ocean: Subsea Dissolved Methane Measurements Using an Embedded Laser Spectrometer Technology. *Environmental Science* & Technology, 52(18), pp. 10543-10551. doi: 10.1021/acs.est.7b06171.
- Jansson, P., Ferré, B., Silyakova, A., Dølven, K. O. & Omstedt, A. 2019. A new numerical model for understanding free and dissolved gas progression toward the atmosphere in aquatic methane seepage systems. *Limnology and Oceanography: Methods*, 0(0). doi: doi:10.1002/lom3.10307.
- Robb, W. 1968. Thin silicone membranes-their permeation properties and some applications. *Annals of the New York Academy of Sciences*, 146(1), pp. 119-137.
- Sommer, S., Schmidt, M. & Linke, P. 2015. Continuous inline mapping of a dissolved methane plume at a blowout site in the Central North Sea UK using a membrane inlet mass spectrometer – Water column stratification impedes immediate methane release into the atmosphere. *Marine and Petroleum Geology*, 68, pp. 766-775. doi: 10.1016/j.marpetgeo.2015.08.020.
- Wankel, S. D., Joye, S. B., Samarkin, V. A., Shah, S. R., Friederich, G., Melas-Kyriazi, J. & Girguis, P. R.
  2010. New constraints on methane fluxes and rates of anaerobic methane oxidation in a Gulf of Mexico brine pool via in situ mass spectrometry. *Deep Sea Research Part II: Topical Studies in Oceanography*, 57(21), pp. 2022-2029. doi: 10.1016/j.dsr2.2010.05.009.